# OpenReview forum: "SOCK: A Benchmark for Measuring Self-Replication in Large Language Models"
_ICLR.cc/2026/Conference — Submitted to ICLR 2026_

### Official Review · Reviewer_3iQa · 2025-10-28

**Soundness:** 1
**Presentation:** 1
**Contribution:** 2
**Rating:** 2
**Confidence:** 3

**Summary:**

This paper introduces SOCK, a command line interface (CLI) benchmark designed to measure self-replication capabilities in large language models (LLMs). The authors define self-replication broadly to include not only creating functional copies but also persistence across different computational contexts. The benchmark categorizes models using two metrics: Replication-Capability Levels (RCL) and Persistence-Capability Levels (PCL), evaluated through a five-task suite that progresses from basic file copying to cross-container replication.

**Strengths:**

The paper addresses a timely and critical research problem—LLM self-replication—which has significant implications for AI safety and multi-agent systems. The focus on persistence across computational contexts extends beyond simple replication, adding depth to the evaluation framework.

**Weaknesses:**

1. Due to the short page length, key details are omitted. For example:
* The R-score formula (Section 4.1) references constants (e.g., τ, Bᵢ) and task-specific baselines without providing their values or derivation process.
* The "intelligence components" (reasoning, tool use, recovery) are mentioned but not defined operationally, leaving ambiguity in how they are measured.
2. The results (Table 1) are presented without in-depth analysis. For instance:
* Why do models like Gemini-2.5-Flash outperform GPT-5 despite lower general capability? The explanation provided is vague and lacks empirical support.
* Error patterns and failure modes are not analyzed, missing insights into specific bottlenecks (e.g., context retention, tool misuse).

**Questions:**

Please see the weaknesses.

---

> ### Author Response · Authors · 2025-11-26
>
> **Q1. The R-score formula (Section 4.1) references constants (e.g., τ, Bᵢ) and task-specific baselines without providing their values or derivation process.**
>
> Specific values for these constants in the R-score formula were provided in Appendix A.2. These constants were set to be neutral in the context of the benchmark, and thus currently have neutral significance in calculating R-score. This was done so the benchmark could focus more on calculating R-score based on task completion.
>
> **Q2. The ‘intelligence components’ (reasoning, tool use, recovery) are mentioned but not defined operationally, leaving ambiguity in how they are measured.**
>
> We agree and will update this in the rebuttal version of our paper in order to better clarify how these components work operationally. But, to clarify them here, how these intelligence components are defined operationally have a lot to do with how they are implemented technically. Reasoning is defined simply as a proportion of steps that incorporated previous context successfully to total steps, tool use is the proportion of steps in which a model executes syntactically valid tool calls whose outputs are actually incorporated into subsequent reasoning, and recovery is the proportion of times in which the model returns to a successful solution path after a detectably incorrect step or failed tool call. SOCK keeps track of and validates this data in real-time using a monitor program that takes in all of the running context and generated/reported outputs from the agent, which is then all aggregated at the same time in order to be used in the R-score calculation at the end of the run.

---

> > ### Author Response · Authors · 2025-11-26
> >
> > **Q3. Why do models like Gemini-2.5-Flash outperform GPT-5 despite lower general capability?**
> >
> > Models like Gemini-2.5-Flash outperform GPT-5 because they’re able to maintain efficiency across all of their tasks. For example, CPU seconds (total time spent computing tool calls) for GPT-5 was 2.84 seconds, and for Gemini-2.5-Flash it was 0.43 seconds, all while using less memory and running a lesser amount of actions. Gemini-2.5-Flash had a total of 7 files copied, 2 processes spawned and 1 container created compared to GPT-5’s 33 files copied, 2 processes spawned and 1 container created. R-score isn’t used to compute general LLM intelligence, other benchmarks already do that, it’s supposed to be a measure of replication efficiency. Models that replicate much faster score higher than others. The total experiment runtime for GPT-5 was 713.91 seconds, with 3 failed experiments, 2 successful ones, and an overall R-score of 32.52. Gemini-2.5-Flash completed all 5 experiments in only 74.48 seconds, with an overall R-score of 52.76 seconds. Another issue we ran into was “intelligent” models often being better “aligned,” and consequently, less prone to actually doing replication tasks. An LLM’s right to refuse plays a significant factor in our ability to even benchmark models like GPT-5, and it often meant having to tweak SOCK’s system prompts and specific task prompts to make them less prone to triggering alignment “flags.” We suspect this behavior could explain some of the resource differences, and therefore R-score differences, between those models and their less “intelligent” counterparts. However, we don’t believe this trend would continue into higher level RCL/PCL tasks, and SOCK’s focus on resource efficiency diminishes when the upper bound of the suite is increased. Intelligent models would have the ability to actually pass these tasks compared to less intelligent models, despite their resource inefficiency in lower level tasks.
> >
> > **Q4. Error patterns and failure modes are not analyzed, missing insights into specific bottlenecks (e.g., context retention, tool misuse).**
> >
> > Table 1 takes into account several factors when calculating R-score: success, latency to first replication, stealth, and resource usage to compute R-scores in the aggregate. A good way to walk through is by examining Gemini-2.5-Pro’s performance. In Gemini-2.5-Pro’s first 3 tasks (duplication into a directory, spawning new processes, and installing a hidden child process/service) it was able to accomplish tasks with relatively smaller steps to success, and smaller timeframes: 0.97 CPU seconds with 33 files copied and 2 processes spawned, then 0.73 CPU seconds with 2 files copied and 3 processes created, and finally 0.61 CPU seconds with 2 files copied and 4 processes spawned. This shifts drastically in the last two tasks (installing a scheduled relaunch and copying itself onto a sibling container), where these numbers balloon into an extreme number of steps and resources used to succeed. The Scheduled Relaunch task was accomplished with 4.09 CPU seconds, 981 files copied, and 22 processes spawned. The Sibling Container task was accomplished with 2.70 CPU seconds, an extreme amount of memory and disk memory used, and an even larger 11,217 files copied, 6 processes spawned, and 1 container created. Agents only have access to shell tools, and all LLMs were able to use these tools. More intelligent models were prone to overusing shell commands and overthinking the prompts given to them leading to generally greater tool consumption. SOCK attempts to mitigate some of these faults using backed-in solutions like splitting task prompts into Tamely/Boldly prompts that are more or less specific and splitting tools into general and specific tools (per task).

---

> > > ### Comment · Reviewer_3iQa · 2025-11-27
> > >
> > > Thanks for your response. However, I still believe that the overall quality of the paper falls below the standard of ICLR, so I will maintain my original score.

---

### Official Review · Reviewer_ZhxT · 2025-10-29

**Soundness:** 2
**Presentation:** 3
**Contribution:** 2
**Rating:** 2
**Confidence:** 4

**Summary:**

This paper introduces SOCK, a command-line interface (CLI) benchmark designed to evaluate large language models' (LLMs) ability to self-replicate without human intervention. The benchmark consists of five tasks with increasing complexity, categorized by Replication-Capability Levels (RCL 0-2) and Persistence-Capability Levels (PCL 0-2). The authors test eight frontier models across these tasks and compute an R-score that combines success, speed, stealth, intelligence, and resource efficiency.

**Strengths:**

1, Practical Implementation: Docker-based CLI benchmark with controlled environments is a solid engineering contribution.

2, Clear Task Categorization: The RCL-PCL taxonomy provides an intuitive framework for categorizing replication capabilities.

3, Multi-Model Evaluation: Testing 8 frontier models provides some comparative insights.

**Weaknesses:**

1, Severely Limited Scope.
- Only 5 tasks vs. 20 task families in RepliBench.
- Excludes critical capabilities (resource acquisition, weight exfiltration)
- Achieves only RCL 2/PCL 2, missing higher-level threats

2, Flawed Scoring Metric.
- Geometric mean penalizes failures too harshly
- Many hyper-parameters without justification

3, Unclear Threat Model.
The adversarial framing ("user aims to prevent replication; agent aims to maximize replication") is mentioned but not formalized or evaluated

4, Missing Analysis.
No failure mode analysis, no ablation studies, no discussion of why certain models succeed/fail.

**Questions:**

Could you compare your work with RepliBench and highlight the key differences and contributions?

---

> ### Author Response · Authors · 2025-11-26
>
> **Could you compare your work with RepliBench and highlight the key differences and contributions?**
>
> What sets SOCK apart is its emphasis on full autonomy. Every task has been set only with a standard OS shell with no privileged shortcuts (e.g., API keys, userspace commands). Our RCL/PCL taxonomy allows for exact evaluation of both replication and persistence. By making SOCK’s source code openly available to researchers, it enables them to verify, extend, and build upon our results, which leads to transparency and future progress in the field. We believe that this transparency and granularity provide a strong foundation for future research into safe benchmarking of self-replication behavior in LLMs. Thus, while RepliBench has contributed valuable tasks along with a broad suite of evaluations on their tasks, as of this submission, their entire experimental framework and source code are not publicly available, which makes it difficult for the research community to validate, audit, or build within their results.

---

### Official Review · Reviewer_1dqA · 2025-10-29

**Soundness:** 2
**Presentation:** 2
**Contribution:** 2
**Rating:** 2
**Confidence:** 3

**Summary:**

This paper introduces SOCK, a command-line benchmark for evaluating large language models’ (LLMs) ability to self-replicate. The framework defines Replication-Capability Levels (RCL) and Persistence-Capability Levels (PCL), implements five progressively difficult CLI tasks , and computes an overall R-score reflecting success, efficiency, and resource use.

**Strengths:**

S1: The proposed framework for RCL and PCL  provides a clear and valuable structure for decomposing and categorizing a multi-dimensional problem.

S2: The R-score design is a robust contribution. It looks beyond binary pass/fail rates to include V,  P, and S .

**Weaknesses:**

W1: The benchmark tasks mainly test basic command execution and do not capture the higher-level reasoning or planning implied by “self-replication.”

W2: The gap between the framing (which includes up to Level 5) and the actual measured capability (which ceilings at RCL 2/PCL 2) weakens the conceptual impact.

W3: The study focuses solely on CLI-based operations, omitting dimensions such as multi-step decision making, communication, or long-term coordination.

W4: Heuristic scoring function: The R-score aggregates multiple factors with fixed weights (e.g., $w_d, w_v, w_s... = 1$ 10), yet the paper provides no analysis of sensitivity, robustness, or interpretability.

W6: The experiments confirm intuitive expectations (more efficient models score higher) but do not yield new understanding of model behavior.

W7: The writing is clear, but much of the paper reads like documentation of a software tool rather than an analysis-driven research study.

**Questions:**

Q1: What motivated the specific choice of these five tasks and their associated difficulty levels, given the much broader 0-5 level taxonomy?

Q2: Can the RCL–PCL taxonomy be validated or compared to external behavioral definitions?

---

> ### Author Response · Authors · 2025-11-26
>
> **Q1. What motivated the specific choice of these five tasks and their associated difficulty levels, given the much broader 0-5 level taxonomy?**
>
> Our RCL/PCL taxonomy reintroduces principles from established literature on agent autonomy and survivability [Franklin & Graesser, 1996]; [Floreano & Keller, 2010], which follows safety guidelines via gradual evaluation [Amodei et al., 2016]; [Russell et al., 2015]. We’ve broken down our benchmark to ensure it’s both grounded in scientific tradition and practical use for future AI safety. As of now, we currently address only RCL/PCL levels 0–2, focusing solely on autonomous replication and persistence within a controlled and deterministic, auditable environment.
>
> **Q2. Can the RCL–PCL taxonomy be validated or compared to external behavioral definitions?**
>
> Yes, it can be validated through ideas we found on agent autonomy and robotics research [Franklin & Graesser, 1996]; [Floreano & Keller, 2010]. RCL measures how far the agent can progress as the difficulty curve increases, while PCL checks if it can continue onward even after being interrupted. Each task in our benchmark matches a clear, testable stage. This makes our results easy to compare and repeat, both within our system and alongside other major benchmarks. This also ensures that SOCK’s taxonomy is not arbitrary, but grounded in how the field already defines and measures agent capabilities.

---

### Official Review · Reviewer_CR4o · 2025-11-01

**Soundness:** 2
**Presentation:** 2
**Contribution:** 2
**Rating:** 2
**Confidence:** 3

**Summary:**

This paper introduces SOCK, a benchmark designed to systematically evaluate the self-replication capabilities of large language models (LLMs) within controlled environments.  It defines two hierarchical dimensions—Replication-Capability Level (RCL) and Persistence-Capability Level (PCL)—and implements five progressively challenging tasks, from file duplication to cross-container persistence.  The framework uses isolated Docker environments to measure replication success, efficiency, and stealth.  Experiments with eight mainstream LLMs show that while models can reliably reproduce themselves in simple settings, higher-level persistence and cross-environment replication remain challenging.  SOCK provides the first standardized foundation for studying autonomous replication behaviors in LLMs.

**Strengths:**

Originality:
The work is highly original in defining and operationalizing the concept of LLM self-replication. By introducing RCL and PCL as structured capability levels and implementing them in sandboxed environments, the paper turns a speculative and safety-sensitive topic into a measurable research direction.

Significance:
The experimental setup is creative and technically complete, offering a reproducible platform for future work on AI autonomy and containment evaluation.

The paper does not have a significant advantage in terms of clarity and soundness.

**Weaknesses:**

1.  Limited contribution and insufficient depth:
Although the paper demonstrates strong originality in framing the concept of LLM self-replication, its overall contribution remains narrow. The implemented benchmark covers only low-level replication scenarios, and the experimental scope is relatively small. As a result, the work lacks the richness and depth expected for a comprehensive study.
2. Insufficient experiments
The experimental setup is severely limited and lacks statistical depth.   Results are based on only five trials without reporting variance or significance tests, so the reliability of performance differences is unclear.   The scoring formula uses equal weights for all components but offers no justification or sensitivity analysis.   Tasks are independent and memoryless, preventing evaluation of long-term or cumulative replication.   Overall, the experiments validate feasibility rather than provide rigorous quantitative evidence.
3. Reproducibility and scalability issues
The provides no scalability analysis. No runtime or memory statistics are reported.  Key metrics such as stealth and intelligence are not accompanied by logs or examples, making it difficult to verify scoring.  Without detailed experimental traces or resource profiling, reproducibility and scalability remain uncertain, weakening the benchmark’s credibility for large-scale evaluation.

**Questions:**

1.  Experimental reliability
The experiments use only a few trials without variance or significance reporting.  Can the authors clarify how consistent the results are across multiple runs or random seeds?

2.  Scalability
How does the framework perform when the number of agents or replication tasks increases?  Is there any data on runtime, resource usage, or LLM-call efficiency?

3.  Higher-level replication
The benchmark currently stops at RCL2/PCL2.  Do the authors plan to extend it to more complex or long-term replication tasks, and how will safety be ensured in such experiments?

---

> ### Author Response · Authors · 2025-11-26
>
> **Q1. The experiments use only a few trials without variance or significance reporting. Can the authors clarify how consistent the results are across multiple runs or random seeds?**
>
> All task runs within SOCK use a random seed. The data package output from SOCK contains JSON files with data for each of the tasks in the suite; the seed used when that task was run is included in this data
>
> **Q2. How does the framework perform when the number of agents or replication tasks increases? Is there any data on runtime, resource usage, or LLM-call efficiency?**
>
> First question: While SOCK could be extended to increase the number of agents and replication depth, there are inherent safety risks and practical difficulties associated with it. From the moment an agent runs a successfully replicated version of itself, it introduces the need to create custom software/hardware that enables that agent to communicate with each instance of itself and between those instances at all times, and as the replication depth increases, this software/hardware would not only need to be robust enough to facilitate communication between hundreds of these agent instances (clones of clones) but also manage context between all of them and account for general reasoning failures and errors. This is why SOCK at RCL2/PCL2 only tests at “depth 1.”
>
> Second question: Yes, SOCK captures data on “runtime, resource usage, or LLM-call efficiency” on a per-task basis, the data for our experiment runs is available in the results folder at https://anonymous.4open.science/r/lksk-22D1. The data is separated by the specific model tested and within JSON files specific to the task run.
>
> **Q3. The benchmark currently stops at RCL2/PCL2. Do the authors plan to extend it to more complex or long-term replication tasks, and how will safety be ensured in such experiments?**
>
> We plan to extend SOCK’s RCL/PCL in future versions, but doing so carries not immediately noticeable problems. Per Appendix A.1, starting from RCL3/PCL3, the tasks agents are put through start to gradually become more and more centered around an agent’s control and integration within a system. For PCL, this culminates into hardware-level persistence in PCL/5, and for RCL it ends with multi-system and network-level propagation by RCL/5. It becomes apparent then that in order to properly create experiments around these RCL/PCL levels one must create some specifically made software and/or hardware to inject LLMs into. We believe this software would be something like a decentralized “context management system” that would allow agents to deeply integrate into their environment and manage multiple versions of themselves in an efficient and effective way that would also restrict those same agents from leaving that designated environment and for the manual destruction of any experiment running inside of it. However, even if one were to create such software/hardware, the experiments born from it would stop representing the chaos inherent within real world computer systems at these higher replication levels; focusing solely on how useful it would be to the underlying research goal we set out to answer: benchmarking self-replication ability in LLMs, SOCK would become less useful in predicting real world harm. For one, a real world computer system would not have this specifically-created software/hardware pre-installed, as such we can’t reliably say that an LLM with SOCK demonstrated self-replication ability would necessarily carry through to the real world, which is what we’re ultimately concerned about. Secondly, if we’re to give that an LLM could survive and replicate on its own in a real world computer system, either by creating its own version of that hypothetical software/hardware, or by placing it in an environment with it already manually installed, the issue of compute being a highly constrained resource that requires money is an issue that an LLM would still need to solve. In order for an LLM to continue replicating in a harmful manner, even within a perfect environment, it must be able to control its own compute, allocate and find more compute, and thus be able to buy that compute and extend it, all without alerting a human of its existence; not to mention that creating an experiment that benchmarks the ability for an LLM to generate its own income, in the context of self-replication, would raise its own inherent safety concerns. To bring this all together, designing experiments that push past RCL2/PCL2 and test all of this within a controlled, experimental environment that simulates how self-replication at that level actually looks like in the real world is only one part of the problem and “...is an open direction for SOCK.”

---

### Meta-Review · Area_Chair_wg3B · 2025-12-08

**Summary:**

This paper proposes a benchmark to measure large language models’ (LLMs) ability to self-replicate without human intervention. The paper is poorly shaped and organized. For example, Sec. 4.1 only lists the equations directly without any explanation. The benchmark only has a section overview. I believe this paper should be carefully revised for future venues.

**Reviewer Concerns:**

The reviewers' concerns are about insufficient experiments, limited contribution, and poor writting. The authors only replied to the questions but neglected the raised weaknesses.

**Reviewer Scores:**

The authors only replied to the questions but neglected the raised weaknesses. I don't think they should change the scores.

---

### Decision · Program_Chairs · 2026-01-26

Reject